# Distinct Genomic Profiles Are Associated with Treatment Response and Survival in Ovarian Cancer

**DOI:** 10.3390/cancers14061511

**Published:** 2022-03-15

**Authors:** Chris J. de Witte, Joachim Kutzera, Arne van Hoeck, Luan Nguyen, Ingrid A. Boere, Mathilde Jalving, Petronella B. Ottevanger, Christa van Schaik-van de Mheen, Marion Stevense, Wigard P. Kloosterman, Ronald P. Zweemer, Edwin Cuppen, Petronella O. Witteveen

**Affiliations:** 1Center for Molecular Medicine and Oncode Institute, University Medical Center Utrecht, Utrecht University, 3584 CG Utrecht, The Netherlands; c.dewitte@hoharuba.com (C.J.d.W.); joachim.kutzera@medizin.uni-leipzig.de (J.K.); a.vanhoeck@umcutrecht.nl (A.v.H.); n.l.nguyen-2@umcutrecht.nl (L.N.); wigard.kloosterman@frametherapeutics.com (W.P.K.); 2Institute of Human Genetics, University Medical Center Leipzig, 04103 Leipzig, Germany; 3Department of Medical Oncology, Erasmus MC Cancer Institute, Erasmus University Medical Center, 3015 GD Rotterdam, The Netherlands; i.boere@erasmusmc.nl; 4Department of Medical Oncology, University Medical Center Groningen, 9713 GZ Groningen, The Netherlands; m.jalving@umcg.nl; 5Department of Medical Oncology, Radboud University Medical Center, 6525 GA Nijmegen, The Netherlands; nelleke.ottevanger@radboudumc.nl; 6Department of Medical Oncology, Meander Medical Center, 3813 TZ Amersfoort, The Netherlands; c.van.schaik@meandermc.nl; 7Department of Medical Oncology, Amphia Hospital, 4818 CK Breda, The Netherlands; mstevense@amphia.nl; 8Department of Gynaecological Oncology, Cancer Center, University Medical Center Utrecht, Utrecht University, 3584 CX Utrecht, The Netherlands; r.zweemer@umcutrecht.nl; 9Hartwig Medical Foundation, 1098 XH Amsterdam, The Netherlands; 10Department of Medical Oncology, Cancer Center, University Medical Center Utrecht, Utrecht University, 3584 CX Utrecht, The Netherlands

**Keywords:** ovarian cancer, whole-genome sequencing, patient stratification, personalized treatment, treatment response

## Abstract

**Simple Summary:**

In most patients with ovarian cancer, their disease eventually becomes resistant to chemotherapy. The timing and type of treatment given are therefore highly important. Currently, treatment choice is mainly based on the subtype of cancer (from a histological point of view), prior response to chemotherapy, and the time it takes for the disease to recur. In this study, we combined complete genome data of the tumor with clinical data to better understand treatment responses. In total, 132 tumor samples were included, all from patients with disease that had spread beyond the primary location. By clustering the samples based on genetic characteristics, we have identified subgroups with distinct response rates and survival outcomes. We suggest that in the future, this data can be used to make more informed treatment choices for individuals with ovarian cancer.

**Abstract:**

The majority of patients with ovarian cancer ultimately develop recurrent chemotherapy-resistant disease. Treatment stratification is mainly based on histological subtype and stage, prior response to platinum-based chemotherapy, and time to recurrent disease. Here, we integrated clinical treatment, treatment response, and survival data with whole-genome sequencing profiles of 132 solid tumor biopsies of metastatic epithelial ovarian cancer to explore genome-informed stratification opportunities. Samples from primary and recurrent disease harbored comparable numbers of single nucleotide variants and structural variants. Mutational signatures represented platinum exposure, homologous recombination deficiency, and aging. Unsupervised hierarchical clustering based on genomic input data identified specific ovarian cancer subgroups, characterized by homologous recombination deficiency, genome stability, and duplications. The clusters exhibited distinct response rates and survival probabilities which could thus potentially be used for genome-informed therapy stratification for more personalized ovarian cancer treatment.

## 1. Introduction

Epithelial ovarian cancer (EOC) is a genetically heterogeneous disease that is characterized by high recurrence rates, the development of chemotherapy resistance, and subsequently, poor survival. Over the past decades, survival rates have hardly improved [1]. Worldwide, yearly ~295,000 women are diagnosed with ovarian cancer, and ~185,000 die of the disease [2]. Primary treatment of advanced disease consists of surgery and platinum-based chemotherapy. The combination of carboplatin and paclitaxel is the most frequently used regimen and most patients respond well to this treatment. However, with every recurrence, the response rates to platinum treatment decline, due to the emergence of chemotherapy resistance. Subsequent disease-free intervals shorten, eventually leading to death from disease in the majority of patients [3,4]. There is an urgent need to find new treatment options, especially for recurrent disease. Patient stratification at recurrence is still mainly based on prior response to platinum treatment and time to recurrent disease, and not on molecular or genetic characteristics [5].

Due to advances in sequencing technologies and their decreasing costs, our understanding of the molecular basis of ovarian cancer has improved. Targeted and whole-exome sequencing of primary high-grade serous ovarian cancer (HGSC) revealed mutations in TP53 in nearly all patients, as well as extensive copy number variation [6,7]. Mutations in BRCA1 and BRCA2 occur in around 20% of patients (due to a combination of germline and somatic events) and are the most prominent cause of defective homologous recombination (HR), which is associated with improved response to platinum-based chemotherapy and PARP inhibition [6,8]. Potential driver genes, other than TP53 and BRCA1/2, are mostly observed in only a small subset of patients, illustrating extensive interpatient tumor heterogeneity [6,7]. Recently, the genomic landscape across different cancers has been studied in unprecedented detail [9,10], and it was shown that clustering based on genomic data can divide patients with the same cancer into clinically meaningful subgroups [11,12,13,14].

Here, we studied the genomic landscape of 132 prospectively collected solid tissue biopsies of 127 patients with metastatic ovarian cancer. Uniquely, our study integrates this data with detailed clinical data including pre- and post-biopsy treatment data, allowing us to analyze the impact of previous treatments on the genome and to evaluate the response to subsequent treatment in relation to specific genomic and mutational characteristics. We identified subgroups of patients that harbored distinct genomic features such as HR deficiency and genomic stability that could benefit from either traditional chemotherapeutics, PARP inhibitors, or other targeted drugs.

## 2. Materials and Methods

### 2.1. Patient Inclusion, Sample Selection and Clinical Data Collection

For this study, we obtained the WGS data of paired tumor and blood control samples from patients with advanced or metastatic ovarian cancer that were included in clinical studies CPCT-02 (NCT0855477) and DRUP (NCT02925234). The Institutional Review Board of the UMC Utrecht and the Netherlands Cancer Institute (IRB UMCU/NCI) approved both studies and all patients provided written informed consent. Part of the cohort described here was previously described (*n* = 95) as part of a pan-cancer analysis paper [10]. Clinical data were obtained from the CPCT-02 and DRUP study registries. Additional clinical data was acquired through a query at the Dutch Cancer Registration (DCR) with the permission of the local CPCT-02/DRUP principal investigators. This concerned data routinely collected for clinical use, including data related to primary disease (date of diagnosis, histopathological diagnosis, FIGO stage, and treatment details), as well as vital status obtained from the population registry and updated till 31 January 2020. Patients with non-epithelial tumors were excluded (*n* = 7), to limit our cohort to EOC. Patients with serous ovarian carcinoma were further classified according to differentiation grade. Poorly (grade 3) and moderately (grade 2) differentiated serous carcinomas were classified as HGSC, while well (grade 1) differentiated serous carcinomas were classified as LGSC. In case the differentiation grade was unknown, we classified serous carcinomas as serous carcinoma not otherwise specified (NOS). Disease status was defined based on treatment history prior to biopsy collection and subsequent treatment. Patients that had received systemic treatment prior to biopsy were regarded as having recurrent disease. For none of the patients that received post-biopsy treatments, this treatment was part of neo-adjuvant chemotherapy. Patients that were treatment-naive prior to biopsy were regarded as having primary disease. For these patients, it was confirmed that subsequent treatment consisted of standard first-line carboplatin and paclitaxel combination treatment. Response to post-biopsy treatment was registered according to the RECIST criteria for solid tumors based on CT imaging and ranged from complete response (CR), partial response (PR), and stable disease (SD) to progressive disease (PD). In one case clinical progression was observed, which was regarded as PD.

### 2.2. WGS and Data Analysis

DNA isolation, WGS of tumor and blood control paired samples, and somatic and germline variant calling (SNV, indel, SV, CNA) were performed using the pipeline of the Hartwig Medical Foundation (HMF-Pipeline, https://github.com/hartwigmedical/pipeline, accessed on 1 February 2021) as described previously by Priestley et al. on solid tumor biopsies with more than 30% tumor purity (according to histopathological assessment) and matching blood samples [10,15,16]. Tumors were sequenced to a median depth of 103× (range 74–137×) and paired germline reference samples to 38× (range 27–54×) (Appendix A, Appendix A). Lower tumor purity scores did not result in a decreased variant detection rate (Appendix A). In contrast to Priestley et al., we kept multiple samples from individual patients. For five patients, two subsequent timepoints of recurrent disease were included (Appendix A, biopsy count). The ploidy value and tumor purity (>20%) for each sample were obtained from the HMF tool PURPLE (https://github.com/hartwigmedical/hmftools/tree/master/purple, accessed on 1 February 2021). For the ploidy fraction, we defined all ranges with a copy number between 1.5 and 2.5 as diploid, the range below as haploid and the range above as polyploid. Clinical tumor purity values were not commonly available from the hospitals and were not used. For the whole genome duplication status, we followed common recommendations as also used in Priestley et al. [10] and based on the bi-nominal distribution of the frequency of autosomal chromosome duplication: A sample is defined as having WGD when the major allele ploidy in more than 10 autosomes is above 1.5.

### 2.3. Gene Mutation Burden Analysis

To find significantly mutated driver genes in our cohort we used the R package dNdSCV [17]. We combined the VCFs for each of the five patients that had two samples. We ran dNdSCV on the whole cohort and also on the primary and recurrent subsets. We used a qglobal_cv cut-off of <0.1.

To identify genes with driver capacities on a per sample basis, we employed the method described in Priestley et al. [10]. Additionally, we extracted germline mutations in *BRCA1* and *BRCA2* with expected pathogenic effects, by selecting frameshift and nonsense variants. Samples with a germline variant in *BRCA1* or *BRCA2* were additionally assessed for loss of heterozygosity. The top 10 genes that were affected in most of our samples were included in the annotation of the cluster plot, as well as *BRCA1* and *BRCA2*. The tumor mutational burden (TMB; mutations per Mb) for each sample was derived by dividing the sum of mutations across the entire genome (SNVs, MNVs, and indels) by the total mappable sequence length of the GRCh37 FASTA file (2858674662) divided by 10⁶, as has been described previously [11].

### 2.4. Mutational Signatures

We counted the occurrence of SNV, indel, and double base substitutions (DBS) contexts that are described at https://cancer.sanger.ac.uk/cosmic/signatures, accessed on 15 May 2020. This included: (i) 96 trinucleotide contexts, which are composed of one of six classes of base substitutions (C > A, C > G, C > T, T > A, T > C, T > G) in combination with the immediate 5′ and 3′ flanking nucleotides, double base substitutions (DBS); (ii) indels stratified by homopolymer length, the presence of flanking repeats or the presence of flanking homology. Additionally, structural variation contexts were extracted by counting translocations, insertions, inversions, deletions, and duplications, categorized below and above 10 Kb.

Mutational signature analysis was performed with the bioconductor R package “MutationalPatterns” [18]. We derived the absolute contribution of the COSMIC v.3 signatures from the mutation context counts for each sample using the “golden ratio search” method. We initially included all COSMIC signatures and ranked them in a top-list by their overall contribution to our samples. We then evaluated which COSMIC signatures contribute consistently to a high number of our samples in runs with fewer signatures from that top-list (Appendix A). We decided on eight SBS, five DBS, and five ID COSMIC signatures that contribute consistently to our samples.

### 2.5. Determining HR-Status

We assessed HR status using the Classifier of HOmologous Recombination Deficiency (CHORD), a random forest model that uses the relative counts of various mutation contexts (primarily deletions with flanking microhomology and 1–100 kb structural duplications) for predicting HR deficiency [19].

### 2.6. Cluster Analysis

As input for the clustering, we used the categorized SNV, DBS, indels and SV counts, and the ploidy fraction. Normalization for each feature was done by mean-centering the values and then dividing them by the standard deviation. Features that are not independent of each other were centered and normalized together instead of feature-wise to preserve individual differences, i.e., the SNVs (C > A, C > G, C > T, T > A, T > C, T > G) were processed together, as well as the indels and the SVs. The distance measure used was (1-Pearson correlation) and hierarchical clustering was performed with the “ward.D” method. To identify the optimal number of clusters, we first attempted to use the elbow method but no clear elbow was visible (Appendix A). We therefore applied bootstrapping using the “pvclust” package [20] to determine at which cluster number stable clusters (*p*-value above 95%) occur (Appendix A). We also performed principal component analysis (PCA) on the data and found that seven principal components explained 95% of the variance (Appendix A). We decided on seven clusters as the optimal number after assessing the results from the bootstrapping and the PCA.

### 2.7. Actionability Analysis

To identify actionable targets per patient, driver genes were explored in three databases: CIViC [21], CGI [22], and OncoKB [23]. Both approved (level A-validated association) and experimental (level B-clinical evidence) therapies were extracted, and a further differentiation was made based on on- and off-label availability for ovarian cancer patients. Drugs available as standard treatments for primary and or recurrent ovarian cancer (platinum agents and PARP inhibitors) were excluded from this actionability analysis.

### 2.8. Statistics

Plotting and statistical analyses were performed using R (software package, version 3.5.2). Kaplan–Meier plots were created using the R package Survival (https://cran.r-project.org/web/packages/survival/, accessed on 8 May 2020). Statistical tests were performed with the Wilcoxon rank-sum test. Correction for multiple testing was performed with the Bonferroni correction.

## 3. Results

### 3.1. Patients with Metastatic Ovarian Cancer Have Diverse Treatment Histories

We analyzed the paired tumor and blood control whole-genome sequencing (WGS) and clinical data of 132 tissue biopsy samples from 127 patients with metastatic ovarian cancer, partially included in the pan-cancer analysis described by Priestley et al. (methods section) [10]. In this study, a tumor biopsy of a patient with metastatic disease was obtained and response to subsequent treatment was monitored as depicted in the flowchart (Figure 1a). The majority of samples were collected at the time of recurrent disease (86%, *n* = 113/132), while a minority was included at the time of primary disease, prior to any systemic treatment (14%, *n* = 19/132) (Table 1, Appendix A). The most common histopathological subtype at time of diagnosis was high-grade serous carcinoma (HGSC; 56%, 74/132) followed by low-grade serous carcinoma (LGSC; 12%, 16/132) and serous carcinoma not otherwise specified (serous NOS; 10%, 13/132). Non-serous subtypes included clear cell carcinoma (*n* = 5) and endometrioid carcinoma (*n* = 5) (Table 1). The majority of tumors were initially diagnosed at advanced disease (FIGO stage III (*n* = 69) and IV (*n* = 40)). Nearly half of the biopsies were taken from metastatic sites in the peritoneum or omentum (48%, 63/132), followed by lymph node (25%, 33/132) and liver metastases (11%, 14/132). The median patient age at the time of biopsy was 63 (range 31–85).

Most patients with recurrent disease had been heavily treated prior to the time of biopsy, and treatment history varied widely between patients (Figure 1 and Appendix A). The median time between diagnosis and study biopsy in patients with recurrent disease was 37 months (including 10 patients with an interval of more than 10 years). Patients were exposed to a mean of 4.1 drugs in this period (range 1–20) and a longer diagnosis-biopsy interval was associated with a higher number of drugs (*p* = 0.01, R^2^ = 0.058) (Figure 1, Appendix A). All patients with recurrent disease were treated with chemotherapeutic agents prior to biopsy (Figure 1b). A subset of the cohort was additionally treated with targeted drugs (34%, 38/113), hormonal therapy (19%, 22/113), and immunotherapy (5%, 6/113). In total, 28 different drugs were administered to the patients in this cohort (Appendix A, Appendix A). Nearly all patients were exposed to the standard first-line treatment for ovarian cancer, carboplatin (110/113, 97%) and paclitaxel (103/113, 91%). Other commonly administered drugs were pegylated liposomal doxorubicin (PLD) (34/113, 30%), bevacizumab (31/113, 27%), gemcitabine (28/113, 25%), and tamoxifen (19/113, 17%). The remaining 22 drugs were distributed to less than 10 patients each. A detailed per-sample treatment history plot is provided for the HGSC patients (Appendix A). The majority of patients received a unique combination of systemic treatments, illustrating the heterogeneity in therapies even in the absence of molecular guidance, and reflecting the clinical challenges in defining optimal treatment strategies for this patient group (Appendix A). A subset of patients was additionally treated with radiotherapy (20%, 23/113) (Appendix A).

### 3.2. The Correlation of Known Clinical Determinants and Drug Response

We evaluated which treatments were administered after the biopsy was obtained, and how patients responded to the given treatment (Figure 2a, Appendix A). In total, 109 patients started a treatment following biopsy collection. Similar to the time period prior to biopsy collection, there was great variety in the treatments administered after biopsy. The most commonly used treatment regimens consisted of carboplatin and paclitaxel ((*n* = 26), including 16 patients with primary disease), carboplatin, gemcitabine and bevacizumab (*n* = 15), and carboplatin and PLD (*n* = 11) (Figure 2a). Response to treatment was monitored radiologically by CT-scanning and assessed according to the RECIST criteria version 1.1 (Figure 2a) [24]. For 95/109 patients (87%), the RECIST response was available, which ranged from complete response (CR 4/95, 4%), partial response (PR 31/95, 33%), and stable disease (SD 43/95, 45%) to progressive disease (PD 17/95, 18%). We confirmed that the group with a favorable response (CR + PR) contained a higher proportion of patients with primary disease compared to the poor response group (SD + PD), 29% vs 13% (Figure 2b) [25]. Further, all tumors that were initially diagnosed as well differentiated exhibited a poor response (Figure 2b).

A shorter platinum-free interval (PFI), defined as the time between the last platinum treatment and the next recurrence, has been associated with a poorer response to subsequent platinum treatment [26]. We investigated whether the PFI correlated with a response based on radiological RECIST assessment in 27 patients [24]. Only two patients experienced a recurrence within six months, of whom one experienced a favorable response to platinum-based (combination) therapy (Figure 2c). Of the patients with a PFI of over six months, 56% (14/25) had a favorable response. Our analysis did not support the clinical prognostic value of the PFI, likely due to a combination of small numbers and a mix of the number of relapses (Appendix A).

### 3.3. Primary and Recurrent Samples Have Comparable Mutational Loads

We subsequently analyzed the WGS data of the 132 EOC tissue biopsies. On average, a total of 10,729 mutations were detected per sample. The total number of mutations did not differ significantly for primary versus recurrent samples (primary mean 10,202 (SD 7548); recurrent mean 10,818 (SD 6108); Wilcoxon rank-sum test *p* = 0.4, Appendix A), in line with recent observations in other tumor types [10]. In contrast, a previous WGS study in ovarian cancer identified a higher mutation frequency in recurrent versus primary samples [7]. Notably, in that study, the majority of samples in the recurrent disease cohort were derived from ascites, while our cohort was entirely made up of solid tumor biopsies. Our primary and recurrent samples contained a comparable number of mutations with high or moderate predicted impact (primary mean 84, SD 72); (recurrent mean 92, SD 55) (Wilcoxon rank-sum test *p* = 0.27, Appendix A).

To detect driver genes in our cohort based on mutational status, we assessed which genes had a higher somatic mutation rate than expected based on the background mutation rate [17]. We identified four genes in this cohort, TP53, KRAS, PIK3CA, and NF1, all known drivers in ovarian cancer (Appendix A) [6,7]. In a sub-analysis restricted to primary samples, only TP53 reached significance, whereas in the recurrent group all four genes were detected. Of note, the power to detect driver genes relies on the sample size which was markedly larger in the recurrent group (*n* = 120) compared to the primary group (*n* = 19).

### 3.4. Copy Number Aberrations Characterize Ovarian Cancer beyond HGSC

HGSC is known for extensive copy number aberrations (CNA). We therefore evaluated CNA on a genome-wide and gene level both across the cohort and per subtype. The average tumor genome ploidy in our cohort was 2.8 (range 1.6–5.9) and the majority of samples (61%, *n* = 80/132) had undergone whole-genome duplication (WGD). This was recently also observed in multiple other types of metastatic cancer [10]. WGD was observed in both primary (68%) and recurrent samples (59%), suggesting that WGD is an early event in tumor evolution, in line with the results of a recent detailed analysis on the evolution of ovarian cancer by Gerstung et al. [27]. There was no difference in the median tumor genome ploidy of samples from primary disease versus samples from recurrent disease (primary 3.1 (SD 1.1), recurrent 2.7 (SD 0.8), Wilcoxon rank-sum test *p* = 0.28, Appendix A). In a cohort-wide analysis, multiple genomic regions containing recurring amplifications and deletions were identified, encompassing copy number gains in genes marked as drivers in ovarian cancer such as MYC and CCNE1 (Appendix A) [9,28]. On average, 412 structural variants (SVs) were detected (range 1–2135), including balanced and unbalanced events. The number of SVs was comparable in the primary disease group (mean 478, SD 495) versus the recurrent disease group (mean 400, SD 338) (Wilcoxon rank-sum test *p* = 0.64, Appendix A).

Next, we evaluated differences in copy number states between the different histological subtypes. As expected, genome-wide copy number aberrations were most distinct in HGSC. Interestingly, the other subtypes also harbored extensive copy number aberrations, although in LGSC samples to a lesser extent (Appendix A). The average non-diploid fraction (combined haploid and polyploid fraction) was 0.58 for HGSC samples, compared to 0.26 for LGSC and 0.38 and 0.37 for endometrioid and clear cell carcinoma samples respectively. Merged CNA profiles per subtype revealed common changes such as a gain of chromosomes 1q and 8q (Appendix A).

### 3.5. HR-Deficient Samples Harbor High TMB and Are Likely to Be HGSC

We have classified 42 samples (33%) as HR deficient using the HR classifier CHORD [19]. That number matches previously published findings [29]. Within CHORD, a further distinction can be made between BRCA1- and BRCA2-subtype HR deficiency. Both subtypes occurred similarly often (20 BRCA1 type vs. 22 BRCA2 type), and both resulted in a significantly higher TMB compared to the HR-proficient samples (Appendix A). While 56% of all samples are diagnosed as HGSC, 86% of the HR-deficient samples have that diagnosis (Appendix A). 

We compared the CHORD results with the available clinical BRCA1/2 genetic test results, which were available for 72 samples. For 60 samples, clinical genetic testing was performed, although no BRCA1/2 hit could be identified. Of those, 15 (25%) were defined as HRD by CHORD. The predicted underlying damaging gene was non-BRCA in 9 cases (ATM, FANCA, FANCM, FANCD2). For 12 samples, a BRCA1 or BRCA2 mutation was identified in the clinic. Of those, 10 were identified as HRD by CHORD (except HMF002316A and HMF000892A). A comprehensive table with all gene hits, TMB and BRCA, status from CHORD, predicted damaging genes and the clinical data can be found in Appendix A.

### 3.6. Mutational Signatures Reflect Treatment History and Tumor Biology

To assess the footprint of biological factors and treatment effects on the tumor genome, we assessed which mutational signatures were present in our cohort. We derived the contribution of COSMIC v.3 mutation profiles to our samples and identified major contributing profiles for eight single bases substitutions (SBS), five double-base substitutions (DBS), and five indel (ID) signatures (Appendix A, Appendix A). The identified signatures (Appendix A) represented mutational processes previously linked to aging (SBS1, 5, 40, DBS2 and 4, ID1, 2, 5, 8), platinum exposure (SBS31 and 35, DBS5), and defective HR (SBS3, ID6). Our findings confirm a recent analysis that identified these mutational signatures in a similar-sized cohort of primary and metastatic ovarian cancer samples [30]. Increased exposure to platinum was associated with higher absolute contributions of the platinum-associated signatures SBS31, SBS35, and DBS5 (Figure 3a–c). Further, samples classified as homologous recombination (HR) deficient according to HR classifier CHORD [19] had a significantly higher contribution of SBS3, ID6, and ID8 (Figure 3d–f). Both ID6 [30] and ID8 have been linked to double-strand break repair through non-homologous end joining (NHEJ) [30]. Double-strand breaks can be induced by radiotherapy [31], though we did not observe an association between ID8 and prior exposure to ionizing radiation therapy nor for other main treatments (Figure 3g and Appendix A). Taken together, the genomic landscape of metastatic EOC is shaped by both endogenous and exogenous clinically relevant mutational processes as multiple mutational processes are simultaneously active in every patient.

### 3.7. Unsupervised Clustering Based on Genomic Input Reveals Seven Distinct Clusters

We subsequently performed unsupervised hierarchical clustering analysis of the entire cohort based on genomic features to identify subgroups with distinct characteristics (Figure 4). Input consisted of sample ploidy (haploid/diploid/polyploid fraction), the number and relative mutation frequencies of single base substitutions (C > A, C > G, C > T, T > A, T > C, T > G), double base substitutions, indels (insertions and deletions in the context of repetitive regions, regions with microhomology and others), and SV categories (deletions, duplications, inversions, insertions, and translocations, divided in long and short events (≥ or <10,000 bp)). PCA analysis on the clustered data identified short structural deletions (<10,000 bp), small deletions with microhomology, and ploidy state as the most important discriminating features of the cohort (Appendix A). We annotated the cluster plot with clinical and genetic data and identified seven distinct clusters (Figure 4 and Appendix A). Bootstrapping analysis revealed a high degree of stability of the clusters (Appendix A). Next, we analyzed the distribution of clinical features across the seven clusters. Primary disease samples did not cluster together but were distributed across six clusters. This is in line with our observation that primary samples are comparable to samples from recurrent disease, regarding average ploidy, WGD status, and the total number of SNVs and SVs. While the main histological subtype, HGSC, was present in all clusters, LGSC samples concentrated in cluster III. No obvious clustering of the other subtypes was observed, likely due to low numbers in these groups. Further, no clustering was observed according to the biopsy site, patient age at biopsy, or disease spread at primary diagnosis (FIGO stage).

### 3.8. Clusters with Specific Genomic and Clinical Features Have Potential Clinical Impact

Assessment of the individual clusters revealed distinct genomic and clinical characteristics with potential prognostic importance. The first two clusters, I and II, were characterized by a large number of mutations and structural variants. Of these, the most prominent were small deletions with flanking microhomology and short structural deletions and duplications (<10,000 bp) (Figure 4 and Appendix A).

On average, the length of structural deletions in clusters I and II (202 and 235 bp) was markedly shorter compared to the remaining clusters (24,473 bp) (*p* = 2.2×10^−16^) (Appendix A). Notably, the HR classifier CHORD independently classified 100% of the samples in clusters I and II as HR deficient based on genome-wide mutation patterns [19]. In contrast, all of the samples in the remaining clusters were classified as HR proficient. *BRCA1-* and *BRCA2-*type HR-deficient samples were randomly distributed across clusters I and II. Biallelic inactivation of *BRCA1* or *BRCA2* that explained the HR-deficiency genotype was observed in 16/42 (38%) samples in cohorts I and II, while 3/42 (7%) samples harbored a single somatic variant. Samples in clusters I and II were among the samples with the highest contribution of mutational signature ID6, which is characterized by deletions with flanking microhomology and associated with HR deficiency (Appendix A). Additionally, SBS3, which has been attributed to HR deficiency, was highly represented in these two clusters (Appendix A). Clusters I and II consisted mainly of HGSC, but also included a poorly differentiated endometrioid and a clear cell carcinoma. The majority of samples in the HR-deficient clusters had an aberration in *TP53* (40/42, 95%), in line with the high number of HGSC samples in these clusters. The main difference between clusters I and II was related to sample ploidy. Cluster II was largely polyploid and all samples had undergone whole-genome duplication (WGD), whereas the average genome ploidy of cluster I was two. Additionally, NF1 aberrations were present in the majority of cluster I (9/17, 53%), while they were rarely observed in cluster II (2/25, 8%). Despite missing data, a trend of improved response to treatment after biopsy was observed in clusters I and II. 17/35 tumors (49%; *n* = 7 missing data) were sensitive to subsequent treatment (CR or PR), versus 18/60 tumors (30%; *n* = 30 missing data) in the remaining cohort (Figure 4). Additionally, survival in the two HR-deficient clusters was among the highest in the cohort. One-year survival from biopsy was 88% (15/17) and 72% (18/25) in clusters I and II, respectively, compared to 52% (47/90) in the remaining cohort, indicating a prognostic advantage for patients within these clusters (Appendix A).

In contrast to clusters I and II, cluster III comprises a genomic stable subgroup (Figure 4). The tumors in this cluster harbor the lowest numbers of SNVs and SVs and have diploid genomes (Appendix A). The majority of samples were wild type for *TP53* (16/20, 80%). Most samples in this cluster were of the LGSC subtype. However, five HGSC were also present in this cluster, of which three harbored a mutation in *TP53*. Genomic stable tumors tend to respond poorly to chemotherapeutic treatment [32]. The overall response to treatment in this cluster was poor (SD/PD 11/14, 79%; *n* = 6 missing data) and the one-year survival from biopsy was 53% (10/19) (Figure 4 and Appendix A).

The distinguishing feature of cluster IV is the abundance of long duplications (Figure 4, Appendix A). Long duplications comprised 51% of SVs in cluster IV compared to 16% in the remaining clusters. A tandem duplicator phenotype (TDP) was previously identified in ovarian cancer and distinguished six subgroups based on tandem duplication span size [33]. The duplication length in cluster IV peaked at around 231 Kb, correlating with TDP group two. TDP group two is characterized by *TP53* mutations and additionally *CCNE1* pathway activation in a third of samples. Likewise, the majority of samples in our duplication cluster IV harbored a *TP53* mutation (10/11, 91%) and in 27% (3/11) the CCNE1 pathway was activated, either through CCNE1 amplification (HMF002954) or a mutation in FBXW7 (HMF001177 and HMF002316) (Appendix A). In contrast, only 7% (8/121) of the samples outside cluster IV had an amplification of CCNE1 and no other FBXW7 mutations were observed.

Cluster VII harbored most indels at repeat regions and most C > T SNVs (Figure 4 and Appendix A). The majority of patients either had a mutation in *TP53* or a mutation in *KRAS* (15/18, 83%). Mutations in *KRAS* and *TP53* were mutually exclusive, indicating distinct underlying tumor biology processes. Notably, across all cohorts, missense mutations in *KRAS* and aberrations of *TP53* were rarely observed simultaneously (2/16, 13%), while amplification of *KRAS* was only observed in the presence of *TP53* aberrations (7/7, 100%). Cluster VII further consisted of a mix of all histological subtypes. The majority of patients in this cluster were treated with platinum-based agents after biopsy and, interestingly, the one-year survival in this cluster was the worst in the cohort (8/18, 44%) (Figure 4 and Appendix A).

Samples in clusters driven by clear mutational patterns such as clusters I and II also show unique mutational signature compositions that are different from the other samples. Other clusters, such as III, V, and VII, contained samples with less unique signature compositions. On average, the signature similarity of samples within the clusters was significantly higher than for the samples in different clusters (Appendix A).

### 3.9. High Frequency of Actionable Targets in the Poor Response Cluster

We assessed whether the genomic data revealed actionable targets in our cohort. We evaluated both level A (approved) and level B (experimental) drugs, for on- and off-label indications. We restricted this analysis to drugs not available as standard of care for patients with metastatic ovarian cancer (excluding platinum agents and PARP inhibitors). In total, 15 actionable genes were identified for which 59 different therapies were potentially available (Appendix A). Nine genes were affected by mutations, three by an amplification and one by a deletion. Moreover, four fusion genes were detected. In 57 samples (43%), an actionable target was identified, yet none of these patients were actually treated accordingly. In nearly all cases it concerned category B off-label drugs, which refers to drugs for which experimental evidence is available based on studies on other cancer types. The genes with a potential actionable target in most people were *KRAS* (hotspot mutation in 16 patients, amplified in seven additional patients), *NF1* (11 patients), and *PIK3CA* (7 patients) (Appendix A). Targetability per cluster varied widely (16–83%) (Figure 5). Despite a high number of duplications in cluster IV, the patients in this cluster did not harbor any targetable amplified genes. Importantly, in the cluster with the worst survival, most patients were treated with platinum-based agents (cluster VII), while this cluster contained the highest fraction of patients with an actionable target, 83% (15/18 samples). Genome-informed treatment stratification might therefore improve the prognosis of the patients in this cluster.

### 3.10. Intrapatient Genomic Stability and Actionability Over Time

For five patients, two subsequent time points during recurrent disease were sequenced. Four pairs were assigned next to each other in the same cluster, indicating that for these patients genomic profiling of recurrent disease was not influenced by the timing of the biopsy (Figure 4, sampleIDs indicated with an asterisk). In line with this, no difference in targetability between the two biopsies of these patients was observed. In contrast, the two biopsies obtained from the fifth patient (HMF000019) had distinct genomic profiles and were clustered apart from each other in clusters II and V, respectively (Figure 4, Appendix A). These biopsies were sampled 10 months apart from distinct tumor locations. Additionally, the mutational profile of both samples showed a completely different pattern. The biopsy obtained at the first time point presented with inactivation of *TP53* and *KMT2C* (HMF000019B), while the second biopsy lacked these mutations and harbored an actionable *KRAS* variant (HMF000019A), indicating the presence of two primary tumors. Finally, we did not identify any cases of BRCA1/2 reversions in the samples with two time points in our cohort [34].

## 4. Discussion

In conclusion, we analyzed WGS data and clinical data of patients with metastatic ovarian cancer. The described cohort is representative of the mixed clinical population of patients with (recurrent) ovarian cancer, including both serous and non-serous histological subtypes and varying treatment histories.

Genomic analysis revealed that primary and recurrent samples have comparable mutational loads. Further, HGSOC samples are characterized by extensive copy number changes including recurrent genes in line with what has previously been reported [6,7]. Simple genomic clustering of this heterogeneous cohort resulted in subgroups with potential clinical impact. We identified seven clusters including two HR-deficiency clusters (I + II), a genomic stable cluster (III), a duplication cluster (IV), and a poor survival cluster (VII) in which clustering-based stratification could potentially improve outcomes. Patients in the HR-deficiency cluster may benefit more from platinum-based chemotherapeutics and PARP inhibitors compared to patients from the genome-stable cluster. Actionability analysis revealed potential targets for treatment, with special promise for cluster VII in which an actionable target was identified in 83% of the patients. Importantly, most of the patients in this cluster were treated with platinum, and one-year survival from biopsy was only 44%. In contrast to a previous hypothesis on a higher rate of actionable amplified genes in samples with a duplication phenotype [33], we did not find evidence to support this claim in cluster IV, possibly due to low numbers.

Previous studies have shown the role of CCNE1 amplification on chemotherapy resistance and poor survival [35,36]. The 15 patients in our cohort with CCNE1 amplification had a tendency of worse survival compared to the other patients, although (and probably due to small sample size) the result was not statistically significant (Appendix A).

We showed that genomic clustering can classify tumors beyond the traditional histopathological classification parameters. While most LGSC samples clustered in the genomic stable cluster (III), two LGSC samples (both wildtype for *TP53* and *KRAS* mutant) ended up in the cluster with the worst prognosis, cluster VII. These samples had few SVs, like the other LGSC samples, but with much more nucleotide variants, with one of them having undergone whole-genome duplication. Furthermore, we identified five HGSC samples in an unexpected cluster, the genomic stable cluster (III). These five samples had lower mutation and SVs counts compared to most HGSC samples. Notably, two out of five samples were wildtype for *TP53*, which, according to updated histopathological guidelines (taking P53 staining into account), might classify them as LGSC. However, the other three samples are likely true HGSC samples, based on the biallelic inactivation of *TP53* in two of them and a *TP53* missense mutation in the other. Identifying these outliers could improve treatment stratification. While (HGSC) samples in cluster III could benefit from a more targeted approach as they might not respond well to standard chemotherapeutics, (LGSC) samples in cluster VII might benefit from early targeted treatment to improve their poor prognosis. These hypotheses should be confirmed in prospective trials with larger cohorts.

Previously, Tothill et al. identified six molecular subtypes of ovarian cancer based on gene expression data, of which four were confirmed as HGSC subtypes in the TCGA dataset [6,37]. The data in our cohort were restricted to WGS and therefore we were not able to correlate our clusters with these transcriptomic subtypes. Integrated WGS and RNA-seq analysis of larger datasets should be conducted to identify overlap between the transcriptomic subtypes and the genomic clusters described here. As the applications of exome and WGS are emerging in the clinic, this opens the way for actionability analysis which can identify therapeutic options for individual patients beyond standard treatment regimens [38,39,40]. Integrating WGS in randomized controlled trials will allow us to evaluate whether treatment allocation can be improved using both histopathological and genomic information. The results from such trials will bring us closer to an individualized treatment approach [41,42], and ultimately, to increasing survival rates for patients with ovarian cancer.

## 5. Conclusions

In-depth analysis of whole-genome sequencing data of ovarian cancers revealed subgroups of tumors with different genomic features which may reflect different functional characteristics. Indeed, we identified different responses to common treatments and survival between subgroups indicating that treatment stratification and/or prognosis of ovarian cancer patients may benefit from tumor genome-derived parameters.

## Figures and Tables

**Figure 1 cancers-14-01511-f001:**
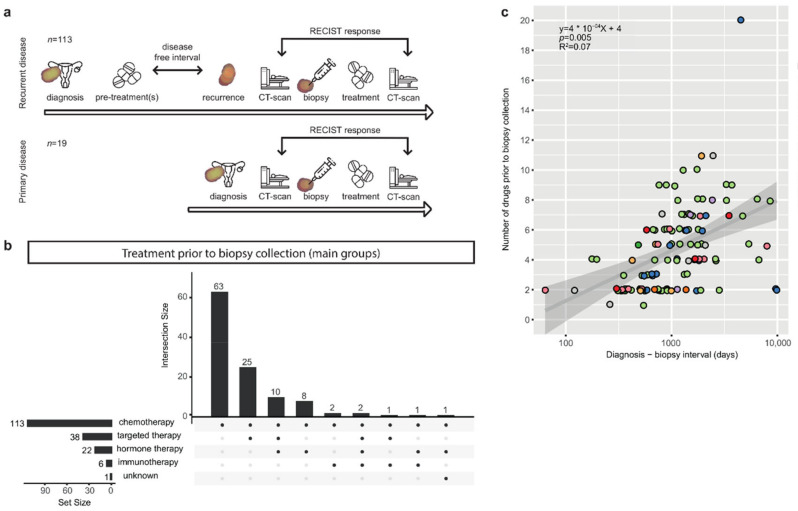
Study flowchart and treatment prior to biopsy. (**a**) Study timeline for biopsy samples obtained at recurrent disease (**top**) and primary disease (**bottom**). (**b**) UpSet plot with treatment history prior to the time of biopsy, categorized by treatment type, for samples that were obtained at recurrent disease (*n* = 113). Horizontal bars (set size) indicate the number of patients that received a treatment type. Vertical bars (intersection size) indicate the number of patients that received a combination of treatment types. All patients with recurrent disease received chemotherapy. (**c**) Scatterplot with diagnosis-biopsy interval in days versus the number of drugs prior to the time of biopsy. NOS = not otherwise specified. A longer diagnosis-biopsy interval was correlated with was associated with a higher number of drugs (*p* = 0.01, R^2^ = 0.058).

**Figure 2 cancers-14-01511-f002:**
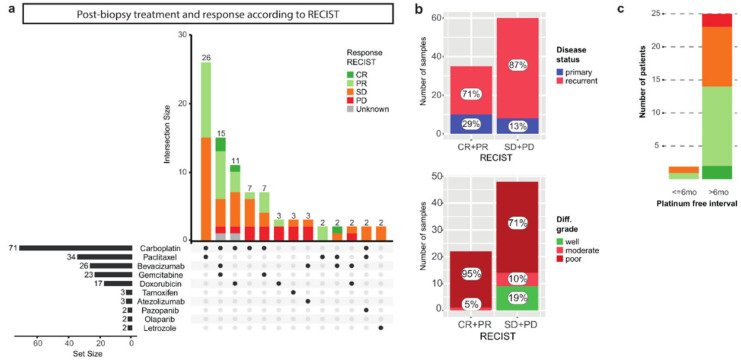
Post-biopsy treatment and RECIST response. (**a**) UpSet plot with post-biopsy treatment. Treatments and treatment combinations given less than twice are cropped from this plot (complete data supplied in Appendix A). Horizontal bars (set size) indicate the number of patients that received a single treatment (including all patients that received a unique combination which is cropped from this plot). Vertical stacked bars (intersection size) indicate the number of patients that received a treatment combination and the response to this treatment combination according to RECIST (version 1.1). CR = complete response, PR = partial response, SD = stable disease, PD = progressive disease. (**b**) Favorable RECIST response (CR + PR) and poor RECIST response (SD + PD) assessed for patients with primary versus recurrent disease, and according to the differentiation grade at diagnosis. (**c**) Response to platinum according to RECIST, for patients with a platinum free interval of less and more than six months. The majority of patients that were re-exposed to platinum had a PFI of more than 6 months. In this group, a favorable response to subsequent platinum treatment was observed in 56% (14/25) of patients.

**Figure 3 cancers-14-01511-f003:**
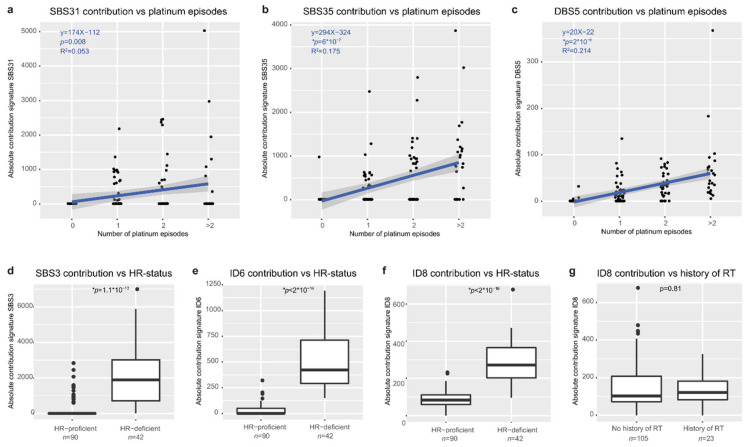
Mutational signatures reflect treatment history and endogenous processes. (**a**–**c**) The absolute contribution of SBS31 (**a**), SBS35 (**b**), and DBS5 (**c**) correlates with increasing platinum exposure. One platinum episode includes multiple platinum cycles. (**d**–**f**) The absolute contribution of SBS3 (**d**), ID6 (**e**), and ID8 (**f**) is significantly increased in patients with HR-deficient tumors. HR deficiency is classified by CHORD. (**g**) The absolute contribution to ID8 is not related to exposure to ionizing radiation therapy (*p* = 0.81). * *p*-value < 0.007 indicated statistical significance (after Bonferroni correction), Wilcoxon signed-rank test.

**Figure 4 cancers-14-01511-f004:**
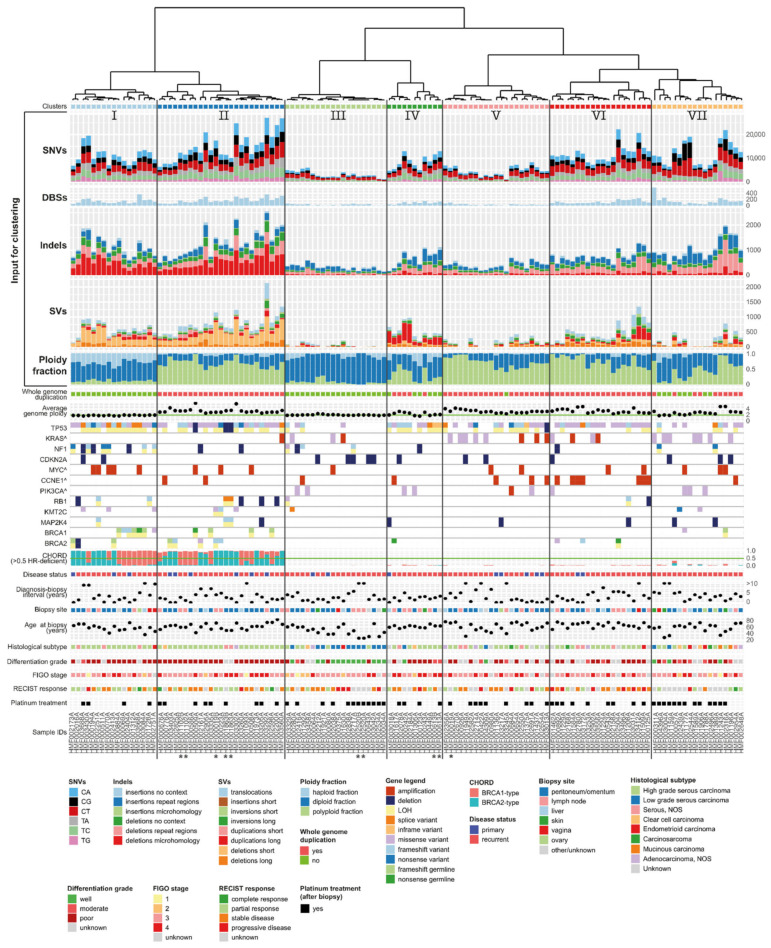
Unsupervised hierarchical clustering based on genomic features reveals distinct subgroups. Dendrogram and cluster plot with seven clusters. Input for clustering consisted of single nucleotide variants (SNVs), dinucleotide variants (DBSs), insertions and deletions (indels), structural variants (SVs), and ploidy fraction (top rows). The plot has been annotated with genetic and clinical data to interpret specific features of each cluster. For oncogenes (indicated by ^) a single row is shown which indicates amplification or a mutation, while tumor suppressor genes are represented by two rows to visualize the effect on both alleles. For five patients, two time points were included (indicated by *), four pairs clustered together.

**Figure 5 cancers-14-01511-f005:**
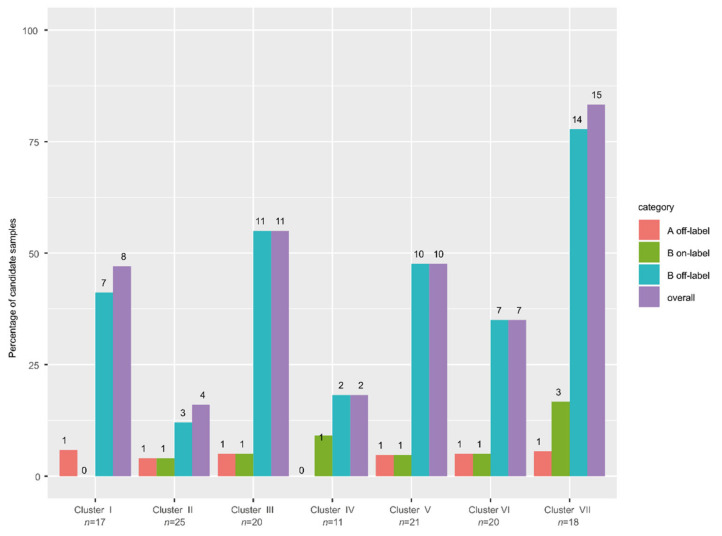
Actionability per cluster. The percentage of samples per cluster with an actionable target (level A/B, on/off label). Purple bar indicates the number of samples per cluster with any actionable target. Most targets are classified as level B off-label. Cluster VII yielded the highest percentage of samples with an actionable target.

**Table 1 cancers-14-01511-t001:** Baseline table with clinical cohort characteristics of 132 patients.

Characteristics	Median/*n*	Range/%
Age at biopsy	63	31–85
Disease status		
primary disease	19	14%
recurrent disease	113	86%
Biopsy Site		
Peritoneum/omentum	63	48%
Lymph node	33	25%
Liver	14	11%
Skin	6	5%
Vagina	5	4%
Ovary	4	3%
Other/unknown	7	5%
Histopathological subtype (at diagnosis)		
High grade serous carcinoma	74	56%
Low grade serous carcinoma	16	12%
Serous carcinoma, NOS ^1^	13	10%
Adenocarcinoma, NOS ^1^	6	5%
Clear cell carcinoma	5	4%
Endometrioid carcinoma	5	4%
Carcinosarcoma	3	2%
Mucinous carcinoma	2	2%
Unknown	8	6%
Differentiation grade (at diagnosis)		
Well	18	14%
Moderate	13	10%
Poor	72	55%
Unknown	29	22%
FIGO stage (at diagnosis)		
I	4	3%
II	9	7%
III	69	52%
IV	40	30%
unknown	10	8%

^1^ NOS: not otherwise specified.

## Data Availability

Genomics and clinical data used in this study is available for free for academic research through an access-controlled mechanism from the Hartwig Medical Foundation. Procedures for requesting access can be found at: https://www.hartwigmedicalfoundation.nl/applying-for-data/, accessed on 23 February 2022.

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
