# Peer review of "Distinct Genomic Profiles Are Associated with Treatment Response and Survival in Ovarian Cancer"

_cancers, 2022, doi:10.3390/cancers14061511_

Round 1

Reviewer 1 Report

This manuscript is written in a very clear and understandable way. Although part of the cohort has already been published in a high-level meta-analysis, the authors have correlated genome-wide sequencing data from tumor material with clinical data and other surveys as new findings, so that new insights have emerged.

My only criticism is based on the fact that only whole genome sequencing data was collected from biopsies of tumor material, and no whole genome sequencing was performed from the germline.

This would make it possible to narrow down the number of mutations, (which are described from line 282 onwards), and to analyze only the pure tumor mutations, and thus possibly evaluate the statistics and ultimately the importance and prognosis of these mutations differently.

It would be desirable if this aspect were mentioned in the discussion.

Author Response

My only criticism is based on the fact that only whole genome sequencing data was collected from biopsies of tumor material, and no whole genome sequencing was performed from the germline.

We apologise for not being sufficiently clear regarding the whole genome sequencing setup. We actually did sequence paired tumor and germline controle samples and could thus efficiently filter for the pure tumor mutations. The depth of sequencing of both samples was indicated in the material and methods, but we now have adapted the methods and results section to better emphasize that tumor-normal pairs were used for the WGS analysis.

Reviewer 2 Report

The study has comprehensively investigated the association of the patient’s clinical responses and outcomes and whole genome profile of a large cohort of ovarian cancers from two clinical studies. While the WGS data of ~90 cases was already included in an early pan-cancer genomic study, significant number of additional cases with detail clinical follow up information is included in the current studies (a total of 132 WGS cases).  Despite the key driver genes and cellular pathways in tumorigenesis were well-reported in previous genome studies, this study has focused on translation of the genome findings for improving current treatment strategies. Through comprehensive bioinformatics analysis, the authors have successfully identified 7 distinct genomic clusters with different responses to various treatment, providing important insights for future personalized clinical management of ovarian cancers. The paper also provides a number of valuable resources for future studies on precision medicine of different ovarian cancer subtypes. It will be interested if the authors can summarize the potential effective treatments for either all or some clusters of ovarian cancer in a conclusion figure

Author Response

there are no comments that need a rebuttal